# Autistic Traits in Schizophrenia: Immune Mechanisms and Inflammatory Biomarkers

**DOI:** 10.3390/ijms26146619

**Published:** 2025-07-10

**Authors:** Maria Suprunowicz, Mateusz Zwierz, Beata Konarzewska, Napoleon Waszkiewicz

**Affiliations:** Department of Psychiatry, Medical University of Bialystok, pl. Wołodyjowskiego 2, 15-272 Białystok, Poland; 39995@student.umb.edu.pl (M.Z.); beata.konarzewska@umb.edu.pl (B.K.); napoleon.waszkiewicz@umb.edu.pl (N.W.)

**Keywords:** schizophrenia, neuroinflammation, inflammation, autistic traits, autistic phenotype, autistic symptoms, Th1, Th2, biomarkers

## Abstract

Autistic traits—such as social communication deficits, cognitive rigidity, and repetitive behaviors—are increasingly recognized in individuals with schizophrenia, particularly in early-onset cases and subtypes with predominant negative symptoms. This overlap has prompted investigations into shared pathophysiological mechanisms. One emerging area of focus is the role of neuroinflammation in schizophrenia, which may contribute to the manifestation of autistic features. Immunological research indicates the presence of chronic low-grade inflammation, microglial activation, and disruption of the blood–brain barrier in schizophrenia. In particular, an imbalance in T-helper (Th) cell responses—specifically a shift toward Th2 dominance or concurrent Th1/Th2 activation—may lead to dysregulated cytokine production and disturbances in neural function. These findings highlight the importance of exploring immunological pathways as a basis for specific symptom profiles. Additionally, current efforts aim to identify reliable inflammatory biomarkers in schizophrenia that could support diagnosis, predict disease course, and guide treatment. Evaluating neuroinflammatory markers in patients with autistic features may provide novel insight into schizophrenia subtypes and help tailor immunomodulatory therapies. This review explores the expression of autistic traits in schizophrenia and examines the role of neuroinflammation and Th1/Th2 imbalance as potential mechanisms and biomarkers.

## 1. Introduction

Mental disorders are diagnosed based on a subjective analysis that aligns with the criteria in classification systems, such as the Diagnostic and Statistical Manual of Mental Disorders (DSM) and the International Statistical Classification of Diseases and Related Health Problems (ICD). The search for biomarkers specific to psychiatry is a key area of research, as it could contribute to the objectification of the diagnostic process, earlier detection of disorders, and more precise tailoring of therapy to individual patient needs [1]. Biomarkers are defined as measurable indicators of biological processes, which may include various substances or parameters indicating the functioning of an organism. They are used to diagnose diseases, predict their course, or evaluate response to treatment [2]. Among the psychiatric conditions that are the subject of intensive biomarker research is schizophrenia. According to the DSM-V, in order to diagnose this disease, a patient must exhibit at least two of the following symptoms: delusions, hallucinations, disorganized speech, significant disorganization or catatonia, and negative symptoms [3]. Current diagnostic criteria may overemphasize positive symptoms while underrepresenting negative and autistic-like features. Moreover, negative symptoms seem to predict the outcome of treatment and precede the onset of psychosis [4]. A distinct group of symptoms, exhibiting greater resemblance to negative than positive symptoms, is characterized as autistic traits. Historically, autism, defined as withdrawal into an internal world, was one of the four main symptoms of schizophrenia [4]. In contemporary psychiatric literature, autism spectrum disorder (ASD) and schizophrenia are regarded as discrete psychiatric entities. ASD is defined as a neurodevelopmental disorder characterized by a pattern of difficulties in social interaction, communication, and stereotypical behavior [5]. Research findings have revealed the presence of a subset of individuals diagnosed with schizophrenia who also exhibit symptoms consistent with autism spectrum disorder [6]. The autistic subgroup of patients diagnosed with schizophrenia exhibits analogous characteristics; however, these characteristics are not as pronounced as those observed in full-blown autism spectrum disorder [5]. The term “schizophrenic autism” has even been introduced, defined as a concept of schizophrenia characterized by dissociation, detachment from reality, emotional indifference, rigidity in attitude and behavior, and dereistic and excessive thinking [7]. Moreover, the concurrence of symptoms as well as the intensity of their simultaneous presentation in individuals diagnosed with both schizophrenia and ASD have been clinically validated as being of considerable significance [8].

Neuroinflammation appears to be a promising direction in the search for biomarkers of autistic traits in schizophrenia. An increasing number of studies indicate that autistic features, such as difficulties in social communication or behavioral rigidity, co-occur in some individuals diagnosed with schizophrenia, especially in the early stages of the disease [6]. At the same time, patients diagnosed with schizophrenia show signs of chronic, low-grade neuroinflammation, including elevated cytokine levels, microglial activation, and blood–brain barrier dysfunction [9,10]. This approach is supported by studies on Th1/Th2 cytokine balance, which indicate an important role of disturbances in this axis in the pathogenesis of the disease [11]. Furthermore, some results suggest that the immune mechanisms associated with this disease may be more complex than assumed by a simple Th1/Th2 imbalance model [11]. Such data highlight the need for further research on the role of the immune system in schizophrenia, which may help identify new therapeutic targets and diagnostic biomarkers. The search for neuroinflammatory biomarkers could, therefore, help not only to better understand the common mechanisms underlying schizophrenia and ASD, but also to identify subtypes of patients who could benefit from targeted immunomodulatory interventions.

## 2. Materials and Methods

This narrative review does not follow a systematic review protocol and has inherent limitations, despite efforts to include a broad range of relevant studies. The literature search was performed using databases such as PubMed, Web of Science, and Google Scholar between October 2024 and June 2025. The search strategy included keywords like “schizophrenia,” “autistic traits,” “autistic symptoms,” “neuroinflammation,” “cytokines,” “inflammation,” and “biomarkers.” Both original research articles and reviews were considered in the initial selection process. Duplicate entries resulting from overlapping search terms were identified and removed using Zotero (version 7.0.14, 64-bit). Abstracts from conference proceedings were excluded based on title and abstract screening. Only studies published in English were included in the final review.

## 3. The Role of Inflammation in the Pathophysiology of Schizophrenia

Chronic, subclinical inflammation plays an important role in the pathophysiology of schizophrenia [12]. One line of evidence supporting the inflammatory hypothesis of schizophrenia stems from neuroimaging studies, which have demonstrated reductions in brain volume among affected individuals [13]. These volume reductions, particularly in structures such as the hippocampus and prefrontal cortex, may result from the action of pro-inflammatory cytokines and neurotoxic processes associated with microgliosis and blood–brain barrier dysfunction [14]. Microglia, the primary immune cells of the central nervous system (CNS), are key mediators of neuroinflammation in schizophrenia [15]. The activation of microglia has been demonstrated to result in the production of pro-inflammatory cytokines, including interleukin-6 (IL-6) and tumor necrosis factor alpha (TNF-α). These cytokines possess the capacity to exert neurotoxic effects. The increased activity of these mediators may contribute to both structural brain damage and the exacerbation of clinical symptoms [16]. Emerging evidence indicates that levels of inflammatory markers vary across different clinical phases of schizophrenia, allowing them to be classified as either state markers—reflecting changes in symptom severity—or trait markers—indicative of chronic disease processes [17]. Alterations in cytokine levels, such as IL-6 and TNF-α, have been correlated with the severity of negative symptoms, cognitive impairment, and structural brain abnormalities [18,19]. Cognitive dysfunction, a core feature of schizophrenia, is closely linked to elevated levels of pro-inflammatory cytokines and oxidative stress. Oxidative stress, driven by an excess of reactive oxygen and nitrogen species, further amplifies neurotoxic mechanisms and impairs neuronal function [20]. The role of inflammation in schizophrenia extends beyond its contribution to disease onset; it also influences the progression and severity of symptoms. This underscores inflammation as a promising target for future therapeutic strategies.

### Th1/Th2 Profile in Schizophrenia

It is known that neurotransmitters and hormones have a significant impact on the functioning of the immune system, regulating processes such as antigen presentation, lymphocyte activity, antibody production, and cytokine secretion [21]. In this context, the division into Th1 and Th2 immune responses is particularly noteworthy. The Th1 response supports the elimination of intracellular pathogens and cancer cells and initiates delayed hypersensitivity reactions. In contrast, the Th2 response supports the fight against extracellular organisms and increases immune tolerance [22]. The Th1 response is associated with the production of pro-inflammatory cytokines such as interferon gamma (IFN-γ), TNF-α, and IL-12 [23]. In contrast, Th2 cells support immune balance by secreting cytokines such as IL-4, IL-5, IL-13, and anti-inflammatory IL-10 [23]. T-lymphocytes can be divided into cytotoxic T-lymphocytes, which are characterized by the CD8 receptor, and helper T-lymphocytes with the CD4 receptor. Activated CD4 T-lymphocytes differentiate into various subgroups of helper T-lymphocytes, namely Th1, Th2, Th17, and regulatory T-cells (Tregs) [24].

Contemporary research indicates that the Th1/Th2 cell profile, which reflects the balance between pro-inflammatory and anti-inflammatory responses, may be a key indicator of the dynamics of inflammation in schizophrenia [12]. Disruption of this balance can lead to the predominance of one type of response, affecting the intensity and nature of the inflammatory process, which potentially modifies both the clinical picture and the course of schizophrenia [25]. Furthermore, studies indicate that the differential activation of microglia and astrocytes, which perform immune cell functions in the central nervous system, plays an important role in the immune imbalance between Th1 and Th2 responses in schizophrenia [26]. Concurrently, the authors emphasize that the findings of studies in this domain exhibit significant inconsistency, thereby complicating the identification of a discernible and replicable pattern [27].

Specifically, Th1-type cytokines, such as IFN-γ, are crucial for initiating pro-inflammatory responses, while Th2-type cytokines—including interleukins IL-4, IL-5, IL-10, and IL-13—serve as regulatory factors that counteract inflammation and maintain immune homeostasis [28,29]. Several researchers have proposed that schizophrenia is associated with an immune imbalance skewed towards a Th2-dominant profile [30,31,32]. This immune shift is characterized by decreased levels of Th1-related cytokines, such as IL-2 and IL-12, alongside elevated levels of Th2-associated cytokines, including IL-4 and IL-10 [31,33]. Chiang et al. demonstrated significantly lower ratios of IFN-γ/IL-4, IFN-γ/IL-10, IL-2/IL-4, and TNF-α/IL-4 in patients with schizophrenia compared to healthy controls, suggesting that Th2 predominance may constitute a distinct immunological feature of schizophrenia [32]. Malashenkova et al. further showed that excessive Th2 cytokine production, particularly IL-4, can inhibit indoleamine 2,3-dioxygenase (IDO)—an enzyme crucial to tryptophan metabolism [33]. Disruption of this metabolic pathway may contribute to N-methyl-D-aspartate (NMDA) receptor hypofunction, which has been implicated in synaptic dysfunction and the development of negative symptoms in schizophrenia [33]. Other studies have also emphasized the link between impaired Th1 responses, reduced IDO activity, and the tryptophan–kynurenine pathway in schizophrenia pathophysiology [26] (Figure 1).

However, the theory of Th2 dominance over Th1 in the pathophysiology of schizophrenia has also been criticized [11]. Criticisms of this theory point to methodological limitations, including analyses based on a restricted set of cytokines. El Kissi et al. reported undetectable IL-4 levels in patients with acute, untreated schizophrenia, suggesting an absence of Th2 activation in the early stages of the illness [34]. Similarly, Fila-Danilow et al. found no association between IL-4 and paranoid schizophrenia in a Polish population [34]. Despite these inconsistencies, the concept of Th1/Th2 imbalance continues to be a relevant direction for immunological research in schizophrenia [35].

Roomruangwong et al. challenged the idea of a clear Th1 or Th2 dominance in schizophrenia, reporting simultaneous activation of both the inflammatory response system (IRS) and the compensatory immune-regulatory reflex system (CIRS) [36]. These results suggest that markers associated with both Th1 and Th2 responses may coexist in schizophrenia, indicating a complex immune phenotype of this disorder. The findings revealed elevated levels of Th1-related cytokines (e.g., IFN-γ, IL-2) alongside increased Th2 cytokines (IL-4, IL-5, IL-13), indicating a complex immune activation pattern [36]. These results are supported by a meta-analysis conducted by Miller et al., which showed concurrent activation of Th1 and Th2 pathways during acute psychotic episodes and no clear dominance of one of the immune response profiles [37]. In this analysis, levels of IFN-γ and IL-12 were elevated, whereas IL-2 remained unchanged [17]. Patient age may also influence Th1/Th2 cytokine profiles, with evidence suggesting a shift toward Th1 dominance with advancing age [32]. Such age-related changes could partly explain discrepancies in study findings that do not account for participant age.

Emerging data also challenge the classical anti-inflammatory classification of Th2 cytokines. In the context of first-episode psychosis (FEP), some Th2 cytokines, such as IL-4, may paradoxically exert pro-inflammatory effects by promoting IFN-γ production [38,39]. Moreover, elevated levels of IL-10 and IL-4 have been correlated with the severity of negative symptoms in FEP patients [40]. Iyer et al. questioned the Th2 dominance model by demonstrating reduced IL-10 levels in both first-episode and recurrent schizophrenia [41]. León-Ortiz et al. also reported conflicting results: while Th1-associated immune activation was observed, increased spontaneous production of IL-4 by peripheral blood mononuclear cells suggested concurrent Th2 activation [42]. These observations support the view that immune dysregulation in schizophrenia involves a more nuanced, multidimensional phenotype.

Within the Th2 profile, increased levels of transforming growth factor-beta (fo-β) and interleukin-1 receptor antagonist (IL-1RA) have been documented, while IL-10 levels remained decreased [37]. Although some meta-analyses have not shown elevated TNF-α levels, several excluded studies reported increases in this pro-inflammatory mediator, which is central to the Th1 response [43].

Zhang et al. observed increased IL-6 concentrations in schizophrenia, without a clear shift toward Th1 or Th2 dominance [44]. Given that IL-6 is produced by various immune cells, including macrophages, this finding highlights the complexity of cytokine signaling beyond the classical Th1/Th2 paradigm. Notably, levels of IL-4 and IL-10 did not differ significantly between patients and controls in this study, suggesting that IL-6 elevation may reflect distinct immune mechanisms unrelated to Th2 dominance [44]. A valuable addition to the analysis of changes in the cytokine profile in schizophrenia is the meta-analysis conducted by Momtazmanesh et al. [45]. While IL-2 levels showed no significant changes in most studies [46,47], results for IFN-γ were mixed—ranging from increased, decreased, or unchanged levels across various cohorts [46,47,48,49,50]. Nevertheless, elevated IFN-γ has been consistently associated with severe symptomatology, supporting its role as a marker of Th1 pathway activation [51]. In addition, studies involving both patients with first-episode psychosis and chronically ill individuals have indicated elevated levels of IL-12, which may suggest its importance in specific phases of the disease [47].

Regarding Th2-related cytokines, elevated levels of IL-5 and IL-13 have been observed in adult patients, suggesting immune activation in specific subgroups [52,53]. In contrast, IL-4 levels generally did not differ between patients and controls [46]. Guo et al. provided important methodological insights, noting that discrepancies in Th1/Th2 balance may depend on the biological sample and assay method [28]. A Th2 shift was observed in serum samples, while in vitro studies revealed Th1 dominance, underscoring the influence of experimental conditions on immune profiling [28]. In vitro research by Chen et al. showed that different classes of antipsychotics can modulate immune cell differentiation in opposing directions. Atypical antipsychotics, such as risperidone, and clozapine, inhibited Th1 differentiation, whereas the typical antipsychotic haloperidol selectively inhibited Th2 differentiation [54].

Some studies also highlight the important role of Treg lymphocytes in the pathogenesis of schizophrenia [55]. The classical Th1/Th2 dichotomy is being reevaluated in light of newly identified T-helper subsets, such as Tregs and Th17 cells. Tregs are associated with TGF-β activity, while Th17 cells secrete IL-17 and IL-10 [11,56]. Dysfunction in these subsets, potentially due to *FOXP3* gene polymorphisms, may destabilize Th1/Th2 balance and disrupt immune homeostasis [57,58]. Reduced numbers of Tregs have been linked to increased production of pro-inflammatory cytokines, reflecting a Th1-skewed inflammatory state [55]. Furthermore, pro-inflammatory cytokines may impair Treg function, exacerbating immune dysregulation in schizophrenia [55].

## 4. The Potential of Th1/Th2 Response as a Biomarker for the Diagnosis of Schizophrenia and in Predicting Response to Treatment

Contemporary research increasingly points to the potential use of the Th1/Th2 ratio as a prognostic biomarker in the context of psychosis diagnosis and response to treatment [59,60]. A review of current research in this area will provide a better understanding of the significance of Th1/Th2 balance in schizophrenia, as well as identify potential directions for further scientific and clinical research. The literature suggests the potential significance of the balance between Th1 and Th2 responses as a biomarker in schizophrenia [61]. The observed decrease in the Th1 to Th2 cytokine ratio may reflect the activation of a Th2-type immune response characteristic of this disease [62]. In one study, IL-33 was identified as a potential biomarker in schizophrenia. Elevated levels of this cytokine, associated with the Th2 immune response, have been found to correlate with the severity of cognitive impairment in patients, making it a promising indicator for assessing clinical status and monitoring disease progression [63]. A study conducted by Noto et al. provides important data on potential biomarkers of schizophrenia, pointing to an intriguing relationship between brain-derived neurotrophic factor (BDNF) levels and immune response markers. The results showed an inverse correlation between BDNF concentration and markers of Th1, Th2, Th17, and Treg lymphocytes [64]. This association suggests that activation of the immune system, regardless of the dominant Th1 or Th2 response profile, may be the mechanism responsible for BDNF reduction in patients experiencing first-episode psychosis [64]. In another study focusing on the acute phase of schizophrenia, significantly elevated C-reactive protein (CRP) concentrations were observed in blood serum. Importantly, this increase showed a positive correlation with the severity of characteristic clinical symptoms of the disease, suggesting a potential link between the severity of the inflammatory process and the intensity of psychopathological symptoms [10]. These results support the hypothesis of a pro-inflammatory nature of the acute phase of schizophrenia, emphasizing the importance of inflammatory markers such as CRP in understanding the pathophysiology of the disorder and their potential use in monitoring disease progression.

Furthermore, the lack of significant differences in IFN-γ levels between the patient group and the control group undermines the hypothesis of a dominant role of the Th1 response in the pathophysiology of schizophrenia [34]. Other studies indicate that IFN-γ and TNF-α may be biomarkers of a weakened cellular immune response in patients with schizophrenia [65]. At the same time, the lack of differences in IL-10 expression undermines its usefulness as a biomarker, suggesting that the humoral response may remain intact in this disorder [65]. The authors suggest that cytokines IL-17 and B-lymphocyte activating factor (BAFF) may be valuable biomarkers for schizophrenia, arguing that their studies showed significantly elevated IL-17 levels and reduced BAFF levels in individuals diagnosed with schizophrenia compared to healthy individuals [34]. Researchers include CCL11 and IL-4 among the biomarkers associated with Th1/Th2 balance [66]. In this context, the authors emphasize that a combination of five biomarkers, including soluble Tumor Necrosis Factor Receptor 1 (sTNF-R1), soluble Tumor Necrosis Factor Receptor 2 (sTNF-R2), C-C Motif Chemokine Ligand 11 (CCL11), Interferon gamma-induced protein 10 (IP-10), and IL-4, may enable the diagnosis of schizophrenia with a sensitivity of 70.0% and a specificity of 89.4% [66].

A meta-analysis of studies on the effects of antipsychotic drugs showed that their use leads to normalization of IL-6 concentrations in patients. These results suggest that IL-6 may be an indicator of disease activity rather than a permanent characteristic [67]. The results of another study suggest that high IL-6 levels at the time of schizophrenia diagnosis may be of prognostic significance. Elevated levels of this cytokine have been associated with an increased risk of developing drug-resistant schizophrenia and longer hospitalizations [60]. These observations point to the potential role of IL-6 as a biomarker for identifying patients with a poorer prognosis, which may support the personalization of therapy and better planning of clinical interventions. Some studies indicate that TNF-α concentration may serve as a marker of predisposition to schizophrenia (trait marker) because its level does not normalize in response to treatment [68]. Similar to TNF-α, IFN-γ is considered a potential trait marker due to the lack of normalization of its level after treatment [67]. Consequently, Dimitrov et al. hypothesize that diminished serum IFN-γ concentrations may signify a diminished Th1 response [69]. In addition, they point to the potential diagnostic value of IL-4 and IL-10 as biomarkers. Elevated IL-4 concentrations in cerebrospinal fluid and increased IL-10 levels in serum and cerebrospinal fluid (CSF) may indicate activation of a Th2-type immune response [69]. The importance of IL-4 as a potential biomarker has also been highlighted in the literature, where reduced levels of this cytokine have been observed in patients with chronic schizophrenia in relapse [47]. In addition, it has been shown that serum IL-4 concentrations correlate positively with the severity of negative symptoms and the presence of depressive symptoms [45]. Studies indicate that elevated levels of IL-2 may potentially serve as a marker associated with negative symptoms and cognitive functioning in patients with schizophrenia [70]. Elevated serum levels of IL-12 appear to be associated with cognitive deficits, indicating its potential significance as a biomarker [45]. In addition, the findings of a study by Boll et al. indicated that leptin, a hormone that plays a pivotal role in regulating the Th1/Th2-dependent immune response, has the potential to serve as a biomarker for schizophrenia. The study also suggested that its dysregulation may be associated with the cognitive deficits and behavioral disorders that are characteristic of this disorder [71]. The literature, including studies by Dunleavy et al., has reported a significant increase in the levels of certain cytokines in patients with a first psychotic episode [72]. However, the authors point out that none of these cytokines meet the criteria for sufficient specificity to be considered a definitive biomarker of FEP [72]. These results highlight the complexity of the immune processes involved in the first psychotic episode and point to the need for further, more detailed research. A comprehensive understanding of the interactions between diverse immune markers is imperative, as is the identification of more precise indicators that could facilitate diagnosis and prognosis. Selected potential markers of neuroinflammation in schizophrenia are presented in Table 1.

However, it is essential to clearly define the clinical utility of the proposed inflammatory biomarkers in schizophrenia, particularly in terms of their potential to inform diagnosis, predict treatment response, or monitor disease progression. Although peripheral blood measurements offer a practical and minimally invasive approach, they present significant limitations when it comes to capturing brain-specific pathological processes. Inflammatory markers detected in circulation may not accurately reflect central nervous system alterations due to the complex interactions between peripheral and central immune mechanisms and the selective permeability of the blood–brain barrier. Therefore, interpreting these biomarkers requires careful consideration of their indirect nature and the methodological challenges inherent in translating peripheral signals into reliable indicators of neuroinflammation.

## 5. Autistic Phenotype in Schizophrenia

The autistic phenotype in schizophrenia refers to the presence of characteristics typical of ASD—mainly social and communication deficits and rigid behavior patterns—in individuals diagnosed with schizophrenia or schizophrenia spectrum disorder, without necessarily meeting the full diagnostic criteria for ASD. The Autism Spectrum Quotient (AQ) scale is used to distinguish this phenotype, with a cutoff score of 35 points [73,74]. However, according to some researchers, this scale is not a sufficiently sensitive screening tool to distinguish between ASD and the autistic phenotype of schizophrenia [75]. Another scale used to detect autistic traits in schizophrenia is the Positive and Negative Syndrome Scale Autism Severity Score (PAUSS)—a special indicator based on eight Positive and Negative Syndrome Scale (PANSS) items such as “blunted affect,” “poor rapport,” “stereotyped thinking,” verified against the Autism Diagnostic Observation Schedule (ADOS) and AQ scales as a practical tool for assessing autistic traits in individuals with schizophrenia spectrum disorders (SSD) [76]. Autistic traits have been shown to occur more frequently in people with bipolar disorder and schizophrenia, regardless of symptom severity, and in depression they are associated with its severity [77]. Researchers suggest similarities between treatment-resistant schizophrenia and ASD. The most prominent cognitive deficits were observed in the treatment-resistant schizophrenia group, followed by the autism spectrum disorder group, with the least prominent cognitive impairments observed in the schizophrenia remission group. Patients with drug-resistant schizophrenia had less pronounced autistic traits as measured by the Autism Quotient than patients with ASD, but more severe than patients in remission [78]. More severe negative symptoms were observed in individuals with autistic traits, which is consistent with their prevalence and reduced social interactions [79]. It also appears that adolescents diagnosed with ASD also exhibit features of the schizophrenia spectrum, which are not limited to negative schizotypal symptoms, but also include disorganized and positive symptoms [80].

Autistic traits are common in individuals with a first episode of psychosis and may be associated with poorer clinical outcomes. At the same time, patients with autistic traits in the study were more likely to be economically inactive and had a diagnosis of mood disorders with psychotic features or a history of brief psychotic disorders [76]. Other studies also confirm that the level of autistic traits is significantly elevated in psychotic disorders and negatively affects social functioning [81]. Furthermore, children who exhibited autistic traits in early life were more likely to develop psychotic experiences in early adolescence [82]. Some studies indicate that autistic traits may co-occur more frequently in patients with schizophrenia who experience symptoms such as polydipsia [83]. Severe autistic traits in early schizophrenia are associated with a significant reduction in social functioning and subjective quality of life, which is why researchers point out that intervention to reduce the severity of these traits may be an important factor in supporting patient recovery [84]. Taking autistic traits into account in the patient’s profile may also be important in therapy, as in the case of mind training (MT), which appears to be significantly less effective in patients with schizophrenia and autistic traits than in patients without these traits [85].

### The Molecular Basis of Autistic Traits in Schizophrenia

The molecular basis of autistic features in schizophrenia remains unknown, although a growing body of research points to the involvement of complex interactions between genetic, epigenetic, and neuroinflammatory processes. It is suspected that autistic traits may result from dysregulation of genes involved in synapse development, neuronal plasticity, and glutamatergic and GABAergic neurotransmission [86]. In addition, a possible role for abnormal microglia activation and chronic low-grade inflammation, which may affect the development of brain structures responsible for social processing, has been observed. There is also evidence that genetic variants common to schizophrenia and autism spectrum disorder, such as mutations in the *SHANK3*, *NRXN1*, and *CNTNAP2* genes, may contribute to the occurrence of autistic phenotypes in some patients with schizophrenia [87,88,89]. Nevertheless, the molecular mechanisms remain largely hypothetical and require further investigation, particularly in the context of identifying biologically coherent subtypes of the disease.

Some studies suggest that the balance of excitation and inhibition, measured by glutamate-glutamine (Glx) and ϒ-aminobutyric acid (GABA) concentrations at rest in vivo using magnetic resonance spectroscopy in the auditory cortex and speech perception centers, is involved in autistic traits and schizotypy [90]. Furthermore, studies indicate that individuals with high glutamate concentrations in the right superior temporal cortex but low GABA concentrations exhibit reduced social and interpersonal skills [91]. Glutamate and GABA abnormalities are reported in autism spectrum disorder, and schizophrenia, and an increased glutamate/GABA+ ratio may affect speech processing and lead to behavioral changes common to both conditions [92]. A study by Sasamoto et al. showed that the severity of autistic traits in individuals with schizophrenia is significantly inversely correlated with the volume of gray matter in the cerebral cortex surrounding the left superior temporal sulcus, a region critical for social information processing [93].

Neuroinflammation may be a promising avenue for the search for biomarkers of autistic traits in the context of schizophrenia. It is increasingly recognized as an important pathophysiological factor in autism spectrum disorder [94]. Studies have observed elevated levels of pro-inflammatory cytokines, microgliosis, and changes in brain immune system function in individuals with ASD [95]. Such changes suggest that inflammatory processes in the central nervous system may be associated with the severity of autistic features. A significant decrease in regulatory lymphocytes, which play a crucial role in maintaining immune homeostasis, has been observed in individuals diagnosed with ASD [96]. Moreover, studies have shown that lymphocytes in patients with ASD exhibit reduced responsiveness to stimulation, abnormal activation, an imbalanced Th/Ts ratio, diminished Th cell activity, and Treg cell deficiency [97]. Both ASD and schizophrenia have been associated with elevated levels of pro-inflammatory cytokines such as IL-6, IL-1β, and TNF-α, as well as microglial activation and altered blood–brain barrier permeability [9,95]. PET studies using ligands for translocator protein (TSPO), a marker of microglial activation, have demonstrated increased neuroinflammatory activity in patients with both ASD and schizophrenia [98]. These findings suggest the possibility of a shared inflammatory basis underlying aspects of the clinical phenotypes of these disorders.

## 6. Conclusions

The presence of autistic traits in some individuals diagnosed with schizophrenia—such as difficulties in social interaction, limited nonverbal communication, and cognitive-behavioral rigidity—may point to a distinct biological foundation. Increasingly, it is recognized that these autistic features in schizophrenia do not fully overlap with either positive or negative symptoms of the disorder. Therefore, classifying these traits as a distinct symptom domain may offer diagnostic and prognostic utility.

Chronic low-grade inflammation, characteristic of schizophrenia, may affect the development and plasticity of neural networks involved in social processing and sensory integration. Elevated levels of pro-inflammatory cytokines and disturbances in microglial activation, as observed in some patients with schizophrenia, are also described in the context of the autism spectrum disorder. These findings suggest that shared neuroimmune mechanisms may underlie both schizophrenia and ASD.

Emerging research highlights the potential relevance of inflammatory biomarkers in schizophrenia for improving diagnostic assessments, predicting disease progression, and guiding therapeutic decisions. Alterations in the Th1/Th2 cytokine balance, such as the decreased Th1/Th2 ratio and elevated levels of Th2-related cytokines like IL-33, have been associated with cognitive dysfunction and symptom severity, suggesting their value as state markers reflecting disease activity. Moreover, markers such as IL-6 and CRP have shown correlations with acute symptom exacerbation and treatment resistance, highlighting their potential role in predicting clinical trajectories and tailoring interventions. Conversely, cytokines including TNF-α and IFN-γ, which do not normalize with antipsychotic therapy, may serve as trait markers, indicating a predisposition to schizophrenia irrespective of current disease state. Although these findings demonstrate promising avenues for clinical application, it is important to recognize that no single biomarker currently offers sufficient specificity or sensitivity to be used as a definitive diagnostic tool. Instead, combining multiple inflammatory markers may improve diagnostic accuracy and risk stratification. Nevertheless, the interpretation of peripheral immune markers warrants caution due to their indirect relationship with central nervous system processes and the complex interplay between peripheral inflammation and neuroinflammation. Further longitudinal studies are needed to validate these biomarkers and integrate them into personalized approaches for the management of schizophrenia.

The identification of neuroinflammatory biomarkers that are specific to autistic traits could, therefore, not only deepen understanding of their pathogenesis, but also enable the separation of this group of symptoms as a separate clinical dimension—alongside positive, negative, and cognitive symptoms.

Neuroinflammation—particularly involving Th1/Th2 imbalance and cytokine dysregulation—may play a key role in the emergence of autistic features in schizophrenia. Current evidence suggests that inflammatory markers may be useful not only for characterizing schizophrenia subtypes but also for identifying individuals with a distinct autistic phenotype. Assessing neuroinflammation as a potential biomarker could enhance diagnostic precision and inform the development of individualized, immunomodulatory treatment strategies. Future research should aim to validate specific neuroimmune signatures associated with autistic traits in schizophrenia and work toward harmonizing diagnostic criteria. Large-scale, longitudinal cohort studies are especially needed to confirm these associations and facilitate the translation of neuroimmune findings into clinical practice. This line of research holds promise for improving patient stratification and developing targeted anti-inflammatory or immunomodulatory interventions.

However, it is worth mentioning that the heterogeneity of methodologies and diagnostic criteria used across the included studies complicates direct comparison and limits the generalizability of conclusions. This review is also constrained by the current state of research, which often lacks large-scale, longitudinal studies examining the direct link between neuroinflammation and autistic traits in schizophrenia. Consequently, some of the discussed associations remain hypothetical and require further empirical validation.

## Figures and Tables

**Figure 1 ijms-26-06619-f001:**
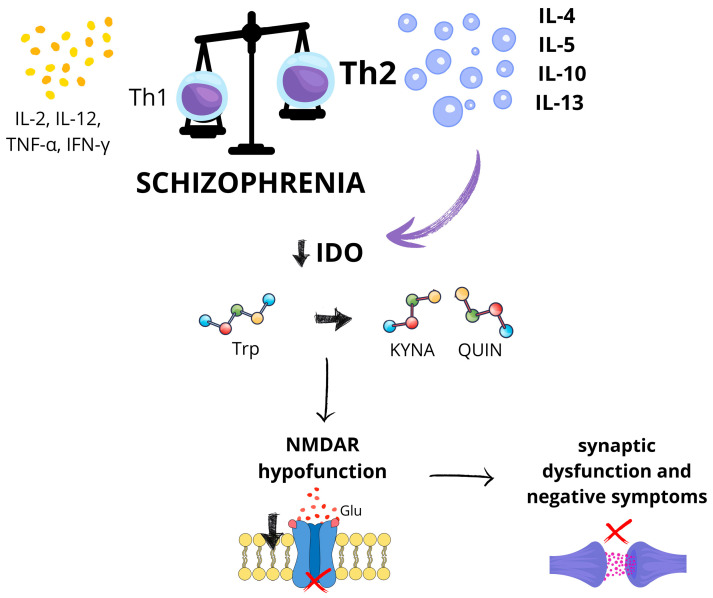
Proposed immuno-metabolic mechanism linking T-helper cell imbalance to N-methyl-D-aspartate receptor dysfunction in schizophrenia. In schizophrenia, the immune system demonstrates a shift from a T-helper 1 (Th1)-type immune response—characterized by the production of pro-inflammatory cytokines such as interleukin-2 (IL-2), interleukin-12 (IL-12), tumor necrosis factor alpha (TNF-α), and interferon gamma (IFN-γ)—toward a T-helper 2 (Th2) response, which includes increased levels of interleukin-4 (IL-4), interleukin-5 (IL-5), interleukin-10 (IL-10), and interleukin-13 (IL-13). This Th2 predominance may suppress the activity of indoleamine 2,3-dioxygenase (IDO), an enzyme responsible for initiating the degradation of the essential amino acid tryptophan along the kynurenine pathway. IDO converts tryptophan into kynurenine, which can then be metabolized into neuroactive metabolites such as kynurenic acid (KYNA) and quinolinic acid (QUIN). Reduced IDO activity may disturb this balance, leading to increased levels of kynurenic acid and/or reduced production of quinolinic acid. This altered ratio may result in hypofunction of the NMDA receptor, impaired glutamate (Glu) signaling, synaptic dysfunction, and the development of negative symptoms in schizophrenia. This figure was generated using Canva; https://www.canva.com (accessed on 3 July 2025).

**Table 1 ijms-26-06619-t001:** Summary of selected immune markers Th1/Th2 response associated with schizophrenia and their clinical relevance: C-reactive protein (CRP), interleukin-4, -6, -12, -17 (IL-4, -6, 12, 17), tumor necrosis factor-alpha (TNF-α), interferon-gamma (IFN-γ), Soluble TNF receptors 1 and 2 (sTNF-R1/2), B-cell activating factor (BAFF), interferon gamma-induced protein 10 (IP-10), C-C motif chemokine ligand 11 (CCL11).

Th1	Th2
⬆ CRP—psychotic symptoms (acute phase) [10] ⬇ IL-6—response to treatment [67], ⬆ IL-6—treatment-resistant schizophrenia, longer hospitalizations [60] TNF-α—treatment-resistance schizophrenia, longer hospitalizations, predisposition marker [68] ⬆ IL-12—negative symptoms and cognition [70], increased cognitive deficits [45] ⬇ IFN-γ—trait marker [67]	⬆ IL-33—exacerbated cognitive dysfunction [63] ⬆ IL-17, ⬇ BAFF—relapse or first-episode psychosis [34] ⬆ sTNF-R1, sTNF-R2, CCL11, and ⬇ IP-10, IL-4—psychosis [66] ⬇ IL-4—chronic schizophrenia in relapse [47], ⬆ IL-4—more negative symptoms and depressive symptoms [45]

⬆—increase in interleukin concentration, ⬇—decrease in interleukin concentration.

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
