# Peer review of "Autistic Traits in Schizophrenia: Immune Mechanisms and Inflammatory Biomarkers"

_ijms, 2025, doi:10.3390/ijms26146619_

Round 1
Reviewer 1 Report
Comments and Suggestions for Authors
Attached are my commnets in the PDF!

Author Response
Dear Reviewer,
We would like to sincerely thank you for the thoughtful and constructive feedback on our manuscript titled “Autistic Traits in Schizophrenia: Immune Mechanisms and Inflammatory Biomarkers”. We greatly appreciate the positive evaluation of the manuscript's relevance, structure, and scientific contribution. Below, we provide detailed responses to each of the comments and describe the revisions made accordingly.
- We have carefully revised the manuscript for clarity, grammar, and language flow across all sections, including those specifically noted by the Reviewer. These changes are now reflected throughout the text.
- All references have been reformatted to strictly follow the IJMS citation style. We ensured consistency in punctuation, bracket use, journal names, volume numbers, and spacing throughout the manuscript and reference list.
- Specific Comments and Suggested Revisions:
- Comment: “This narrative review discusses…” could be revised for flow.
Response: Revised as suggested. The sentence now reads: “This review explores the expression of autistic traits in schizophrenia and examines the role of neuroinflammation and Th1/Th2 imbalance as potential mechanisms and biomarkers.” - Comment: Line 44: “the diagnosis places excessive emphasis…” Response: Revised for clarity and precision. The sentence now reads: “Current diagnostic criteria may overemphasize positive symptoms while underrepresenting negative and autistic-like features.”
- Comment: Consistently refer to “autistic traits” or “autistic features” rather than alternating between terms such as “autistic symptoms” and “schizophrenic autism.” Response: We have reviewed and revised the terminology throughout the manuscript. We now consistently use the term “autistic traits” or "features" to ensure clarity and coherence in terminology.
- Comment: Line 443–458: This section is strong in content but would benefit from language polishing. Response: The indicated section has been carefully edited for grammar, conciseness, and clarity. The revised sentence now reads: “Classifying these traits as a distinct symptom domain may offer diagnostic and prognostic utility.”
- Comment: Line 469–474: Consider offering a brief suggestion for addressing these limitations in future research. Response: Sentences have been added: “Future research should aim to validate specific neuroimmune signatures associated with autistic traits in schizophrenia and work toward harmonizing diagnostic criteria. Large-scale, longitudinal cohort studies are especially needed to confirm these associations and facilitate the translation of neuroimmune findings into clinical practice. This line of research holds promise for improving patient stratification and developing targeted anti-inflammatory or immunomodulatory interventions.”
- Comment: Please ensure all references are formatted according to IJMS style. Response: Completed. All references have been updated to comply with IJMS formatting requirements, and inconsistencies have been corrected.
- Comment: Several long sentences throughout the manuscript would benefit from restructuring. Response: We have revised multiple long sentences throughout the manuscript to improve readability and flow. The specific sentence noted by the Reviewer now reads: “These findings suggest that shared neuroimmune mechanisms may underlie both schizophrenia and ASD.”
We are grateful for your insightful feedback, which helped us improve the quality and clarity of our manuscript. We believe the revisions made in response to your comments have significantly strengthened the paper. Likewise, we hope the current version meets the standards for publication in International Journal of Molecular Sciences.
Thank you once again for your thoughtful review and recommendation.
Sincerely,
The Authors

Reviewer 2 Report
Comments and Suggestions for Authors
Review of Article: Autistic Traits in Schizophrenia: Immune Mechanisms and Inflammatory Biomarkers
Introduction and Overview
The reviewed article presents a thorough examination of autistic traits observed among individuals diagnosed with schizophrenia, emphasizing their prevalence in early-onset cases and those characterized predominantly by negative symptoms. The authors address the increasing recognition of overlaps between these two conditions and explore the underlying pathophysiology, particularly focusing on neuroinflammation and immune-mediated processes.
Strengths
Comprehensive Literature Review: The article incorporates extensive bibliographic sources, including recent studies within the last five years, reflecting state-of-the-art developments in the field.
Logical Structure: The manuscript follows a logical sequence from introduction to discussion, guiding readers through the multifaceted relationship between autistic traits and schizophrenia.
Integration of Multidisciplinary Perspectives: Neuroscience, immunology, and clinical psychiatry converge in the exploration of autistic-like symptoms in schizophrenic populations.
Identification of Knowledge Gaps: Clear identification of unresolved questions invites further research.
Weaknesses
Limited Original Research: As a narrative review, the article relies heavily on pre-existing studies, limiting direct contributions beyond synthesis.
Potential Overemphasis on Cytokine Imbalances: Some sections risk overstating the role of cytokine levels relative to other contributing factors.
Visual Representation Needs Improvement: Enhancing visual elements (tables, diagrams) could facilitate better understanding of complex immunological pathways.
Recommendations for Improvement
To elevate the impact of this review, consider the following suggestions:
Include concise graphic representations illustrating key relationships (e.g., cytokine signaling cascades).
Address the clinical applicability of proposed inflammatory biomarkers explicitly.
Emphasize challenges associated with measuring brain-specific changes indirectly using peripheral blood assays.
Conclusion
Despite being a narrative review, this work makes significant strides towards integrating disparate bodies of knowledge. Through careful evaluation of the literature, the authors lay a foundation for targeted diagnostic approaches and tailored immunomodulatory therapies. Future studies building on this framework promise substantial advancements in understanding the intersection of schizophrenia and autism-related phenotypes.
Final Verdict: An excellent review article deserving publication after addressing suggested minor revisions.
Author Response
Dear Reviewer,
We sincerely thank you for the detailed and thoughtful evaluation of our manuscript titled “Autistic Traits in Schizophrenia: Immune Mechanisms and Inflammatory Biomarkers.” We are grateful for the recognition of the article’s strengths, including the comprehensive literature review, logical structure, and multidisciplinary integration. Likewise, we greatly appreciate your constructive comments and have addressed each of them as detailed below.
Reviewer’s Comments and Our Responses
- Comment: “Limited Original Research: As a narrative review, the article relies heavily on pre-existing studies, limiting direct contributions beyond synthesis.” Response: We acknowledge this limitation, inherent to the narrative review format. To add more value, we have further emphasized critical gaps in current research.
- Comment: “Potential Overemphasis on Cytokine Imbalances: Some sections risk overstating the role of cytokine levels relative to other contributing factors.” Response: We appreciate this observation. To maintain a balanced perspective, we have revised relevant sections to clarify that cytokine dysregulation represents one of several interrelated mechanisms contributing to the pathophysiology of autistic traits in schizophrenia.
- Comment: “Visual Representation Needs Improvement: Enhancing visual elements (tables, diagrams) could facilitate better understanding of complex immunological pathways.” Response: In response to this valuable suggestion, we have improved the clarity and labeling of the figure included in the manuscript. A revised version of the graphical summary depicting Th1/Th2 imbalance, IDO pathway modulation, and NMDA receptor hypofunction has been uploaded to better visualize the proposed mechanisms. Additional explanatory text has also been added to the figure legend.
- Suggestion: “Include concise graphic representations illustrating key relationships (e.g., cytokine signaling cascades).” Response: As noted above, we have updated the figure to visually summarize the Th1/Th2 shift and its downstream effects, including IDO inhibition and NMDA receptor hypofunction. This revised figure is now included in the manuscript with an expanded legend for clarity.
- Suggestion: “Address the clinical applicability of proposed inflammatory biomarkers explicitly.” Response: We have added a short paragraph in the Conclusions section (612-630) explicitly addressing the clinical implications of using inflammatory biomarkers for the diagnosis and subtyping of schizophrenia with autistic traits.
- Suggestion: “Emphasize challenges associated with measuring brain-specific changes indirectly using peripheral blood assays.” Response: New sentences have been added (489-499) acknowledging the inherent challenge of extrapolating CNS-specific immune activity from peripheral cytokine profiles.
We are grateful for your positive feedback and helpful suggestions. We believe the revisions made in response to your comments have improved the clarity, clinical relevance, and overall quality of the manuscript. Thank you again for your time and expert review.
Sincerely,
The Authors
